# Minimalistic In Vitro Culture to Drive Human Naive B Cell Differentiation into Antibody-Secreting Cells

**DOI:** 10.3390/cells10051183

**Published:** 2021-05-12

**Authors:** Peter-Paul A. Unger, Niels J. M. Verstegen, Casper Marsman, Tineke Jorritsma, Theo Rispens, Anja ten Brinke, S. Marieke van Ham

**Affiliations:** 1Department of Immunopathology, Sanquin Research and Landsteiner Laboratory, Amsterdam UMC, University of Amsterdam, 1066 CX Amsterdam, The Netherlands; peterpaul.unger@planet.nl (P.-P.A.U.); n.verstegen@sanquin.nl (N.J.M.V.); C.Marsman@sanquin.nl (C.M.); T.Jorritsma@sanquin.nl (T.J.); T.Rispens@sanquin.nl (T.R.); a.tenbrinke@sanquin.nl (A.t.B.); 2Synthetic Systems Biology and Nuclear Organization, Swammerdam Institute for Life Sciences, University of Amsterdam, 1098 XH Amsterdam, The Netherlands; 3Swammerdam Institute for Life Sciences, University of Amsterdam, 1098 XH Amsterdam, The Netherlands

**Keywords:** costimulatory molecules, cytokines, cell differentiation, transcription factors

## Abstract

High-affinity antibody-secreting cells (ASC) arise from terminal differentiation of B-cells after coordinated interactions with T follicular helper (Tfh) cells in germinal centers (GC). Elucidation of cues promoting human naive B-cells to progress into ASCs is challenging, as this process is notoriously difficult to induce in vitro while maintaining enough cell numbers to investigate the differentiation route(s). Here, we describe a minimalistic in vitro culture system that supports efficient differentiation of human naive B-cells into antibody-secreting cells. Upon initial stimulations, the interplay between level of CD40 costimulation and the Tfh cell-associated cytokines IL-21 and IL-4 determined the magnitude of B-cell expansion, immunoglobulin class-switching and expression of ASC regulator *PRDM1*. In contrast, the B-cell-specific transcriptional program was maintained, and efficient ASC formation was hampered. Renewed CD40 costimulation and Tfh cytokines exposure induced rapid secondary STAT3 signaling and extensive ASC differentiation, accompanied by repression of B-cell identity factors *PAX5, BACH2* and *IRF8* and further induction of *PRDM1.* Our work shows that, like in vivo, renewed CD40L costimulation also induces efficient terminal ASC differentiation after initial B-cell expansion in vitro. This culture system for efficient differentiation of human naive B-cells into ASCs, while also maintaining high cell numbers, may form an important tool in dissecting human naive B-cell differentiation, thereby enabling identification of novel transcriptional regulators and biomarkers for desired and detrimental antibody formation in humans.

## 1. Introduction

Production of high-affinity antibodies by long-lived plasma cells (PC) and the generation of long-lived memory B-cells (MBC) are essential characteristics of T-cell-dependent humoral immunity to combat invading pathogens and protect upon pathogen re-encounter. On the downside, undesired antibody responses result in pathologies, such as alloimmunity [1,2] and auto-immunity [3,4,5,6]. Generation of class-switched high-affinity PCs and MBCs upon antigen encounter of B-cells also requires CD40 costimulation and cytokines, like IL-21 and IL-4, provided by T follicular helper (Tfh) cells in germinal center (GC) reactions in secondary lymphoid organs [7,8,9,10,11,12,13]. A GC contains two distinct morphological regions, the dark zone (DZ) and the light zone (LZ). In the DZ, GC B-cells (GCBC) undergo expansion and hypermutation of the genes that encode the immunoglobulin V region that affect antigen binding. GCBC that acquired B-cell receptor (BCR) mutants with increased affinity preferentially bind intact antigen displayed on the surface of follicular dendritic cells (FDC) in the LZ, which results in antigen internalization and presentation of antigen-derived peptides via major histocompatibility complex class II molecules (pMHCII) on the B-cell surface [14]. Subsequently, Tfh cells in the LZ engage in cognate interaction with GCBC necessary for GCBC selection. GCBC frequently migrate between DZ/LZ, which ultimately leads to the differentiation of GCBC with the highest affinity into the effector populations [15]. In vivo, GCBC thus repeatedly receive Tfh signals before terminal differentiation into antibody-secreting cells (ASCs), which consist of short-lived plasmablasts (PB) and long-lived PCs.

Insight into the factors that determine the actual fate decision of GCBC to become either MBC or ASC is being gained. Antibody-mediated inhibition and genetically manipulated mouse models have demonstrated an essential requirement for CD40L [16,17] and IL-21 signaling [18,19], but also a contribution of inducible T cell costimulatory (ICOSL) [20], complement receptor (CR) 1 and 2 [21], programmed cell death protein 1 (PD1) and its ligands [22], and CD80 [23] in long-lived PC fate decision have been indicated. CD40 stimulation of naive B-cells combined with cytokines induces efficient proliferation, isotype switching and formation of a CD27^+^ B-cell population in vitro. In contrast, ASC formation is only observed in cultures with very limited B-cell numbers, which do not allow elucidation of the ASC differentiation process [10,24]. This demonstrates that the current in vitro culture systems do not reflect the in vivo situation well enough. Previous research has demonstrated that strong CD40 signaling facilitates stable GCBC:Tfh cell interactions to drive the generation of ASC [25]. In addition, graded levels of CD40-CD40L interactions in total B-cells regulated expansion and differentiation in the presence of IL-2, IL-4 and IL-10 in vitro [26]. However, this study did not elucidate the effects of CD40 costimulation on naive B-cell differentiation.

An in vitro system that allows efficient naive B-cell differentiation into the ASCs is highly desired to (i) unravel the process(es) involved in ASC formation, (ii) identify the factors that control the fate decision of differentiating B-cells into ASCs and find targets to modulate antibody formation in vivo and (iii) identify membrane markers associated with the distinct stages of B-cell to ASC formation with diagnostic potential for (early) assessment of undesired short-lived or long-lived antibody formation. Therefore, we established an in vitro culture system for human naive B-cell differentiation into ASCs by varying the level and frequency of CD40 costimulation. We demonstrate that high CD40 costimulation and IL-21 signaling induce naive B-cell expansion but do not allow full ASC formation due to maintenance of B-cell-specific transcription factors, *PAX5, IRF8* and *BACH2*. Renewed CD40 costimulation and IL-21 allowed effective ASC differentiation by extinguishing the B-cell transcriptional program, which correlated with the re-induction of STAT3 signaling.

## 2. Materials and Methods

### 2.1. Cell Lines

NIH3T3 fibroblast cells (3T3) were stably transfected with Fsp I linearized human CD40L plasmid (a kind gift from G. Freeman) [27,28] and Pvu I linearized pcDNA3-neomycin plasmid in 3T3 medium (IMDM (Lonza, Basel, Switzerland) containing 10% FCS (Bodinco, Alkmaar, the Netherlands), 100 U/mL penicillin (Invitrogen, Carlsbad, CA, USA), 100 μg/mL streptomycin (Invitrogen, Carlsbad, CA, USA), 2 mM L-glutamine (Invitrogen, Carlsbad, CA, USA) and 50 μM β-mercaptoethanol (Sigma Aldrich, St. Louis, MO, USA) using the Lipofectamine 2000 DNA transfection reagent protocol (Invitrogen, Carlsbad, CA, USA). After three days, transfectants were cultured in 3T3 medium supplemented with 500 µg/mL G418 (Life Technologies, Carlsbad, CA, USA). Transfectants were sorted four times based on their CD40L expression using flow cytometry (anti-CD154; clone TRAP1; BD Bioscience) to select subclones stably expressing low (CD40L^+^) and intermediate (CD40L^++^) CD40L levels than the already available NIH3T3 fibroblasts expressed high (CD40L^+++^) human CD40L levels [27]. Comparison of CD40L levels between the 3T3 cells expressing low, intermediate and high CD40L levels was done via flow cytometry after trypsinization. These cells were also cultured in 3T3 medium supplemented with 500 µg/mL G418.

### 2.2. Isolation of B Cells from Human Healthy Donors

Buffy coats were collected from voluntary, non-remunerated, adult healthy blood donors (Sanquin Blood Supply, Amsterdam, the Netherlands), who provided written informed consent for the use of remainders of their donation for research as part of routine donor selection and blood collection procedures. Peripheral blood mononucleated cells (PBMCs) were isolated from buffy coats using a Lymphoprep (Axis-Shield PoC AS, Dundee, Scotland) density gradient. Afterward, CD19^+^ B-cells were separated using magnetic anti-CD19 Dynabeads and DETACHaBEAD (Invitrogen, Carlsbad, CA, USA) according to manufacturer’s instructions with purity >99%.

### 2.3. In Vitro Naïve B Cell Differentiation Cultures

3T3 WT and/or CD40L-expressing 3T3 cells were harvested and irradiated with 30 Gy. Equal numbers (1 × 10^4^) WT or CD40L-expressing (+, ++ or +++) 3T3 cells or, alternatively, 9:1 WT to CD40L^+++^-expressing 3T3 cell ratio were seeded in B-cell medium (RPMI 1640 (Gibco, Dublin, Ireland) without phenol red containing 5% FCS, 100 U/mL penicillin, 100 μg/mL streptomycin, 2mM L-glutamine, 50 μM β-mercaptoethanol and 20 μg/mL human apotransferrin (Sigma Aldrich, St. Louis, MO, USA; depleted for human IgG with protein G sepharose)) in 96-well flat-bottom Nunc plates (Thermo Fisher Scientific, Waltham, MA, USA) to allow adherence overnight.

The next day, CD19^+^CD27^−^IgG^−^ naive B-cells, to prevent BCR-mediated activation, were sorted on a FACSAria II. To assess proliferation, FACS-sorted naive B-cells were labeled with 0.5 μM CFSE (Invitrogen, Carlsbad, CA, USA) in PBS for 15 min at room temperature. Labeling was stopped by adding B-cell medium. Immediately after labeling, 2.5 × 10^4^ naive B-cells were cultured on the irradiated CD40L-expressing 3T3 fibroblasts in the presence of optimal concentrations (data not shown) of IL-21 (50 ng/mL; Invitrogen, Carlsbad, CA, USA) and/or IL-4 (100 ng/mL; CellGenix, Freiburg im Breisgau, Germany) for six and eleven days, without refreshing medium during these cultures, to assess IgG B-cell formation, antibody-secreting cell (ASC) differentiation and Ig secretion in the culture supernatants.

For secondary cultures, six-day stimulated B-cells were collected, washed and 2.5 × 10^4^ cells were re-cultured in fresh medium on irradiated CD40L-expressing 3T3 fibroblasts (as described above), including cytokines as in the primary cultures for 5 days.

B-cell receptor stimulations were performed as previously described using soluble anti-IgM antibodies or anti-IgM-coated 3 μm polystyrene beads (Spherotech, Lake Forest, IL, USA) [29]. For soluble antibody stimulation, B-cells were first incubated with 10 µg/mL mouse anti-human IgM (clone MH15-1; Sanquin Reagents, Amsterdam, the Netherlands) for 15 min. Excess non-bound antibodies were washed, and conditions were incubated with 10 μg/mL rat anti-mouse IgG1 antibodies for 15 min to crosslink all bound anti-IgM. Excess non-bound antibodies were washed, and 2.5 × 10^4^ naive B-cells were cultured on irradiated CD40L-expressing 3T3 fibroblasts as described above. For stimulation with anti-IgM-coated beads (coated according to manufacturer’s instructions), B-cells were pre-incubated with beads in a B:bead ratio of 1:2 for 30 min at 37 °C before putting the cells in co-culture with the irradiated CD40L-expressing 3T3 fibroblasts as described above.

### 2.4. Flow Cytometry

Cells were first washed with PBS and stained with LIVE/DEAD fixable near-IR (dead cell stain kit, Invitrogen, Carlsbad, CA, USA) for 30 min at room temperature in the dark. Then, cells were washed with PBS supplemented with 1% bovine serum albumin. Extracellular staining was performed by incubating the cells for 30 min at room temperature in the dark with the following antibodies: Anti-CD19 (clone SJ25-C1, BD Bioscience, Franklin Lakes, NJ, USA), anti-CD20 (clone L27, BD Bioscience, Franklin Lakes, NJ, USA), anti-CD27 (clone L128, BD Bioscience, Franklin Lakes, NJ, USA; clone O323, eBioscience, San Diego, CA, USA), anti-CD38 (clone HB7, BD Bioscience, Franklin Lakes, NJ, USA) and anti-IgG (clone G18-145, BD Bioscience, Franklin Lakes, NJ, USA or clone MH16-1, Sanquin Reagents, Amsterdam, the Netherlands). Samples were measured on a FACSCanto, FACSLSRII or FACSLSR Fortessa (BD Bioscience, Franklin Lakes, NJ, USA) and analyzed using FlowJo software version 10 (Treestar, Ashland, OR, USA).

### 2.5. Analysis of STAT Phosphorylation

A total of 2 × 10^5^ naive B-cells were cultured with 0.8 × 10^5^ CD40L^+++^-expressing 3T3 cells in a 24-well Nunc plate (Thermo Fisher Scientific, Waltham, MA, USA) in the absence or presence of IL-21 for 6, 36, 72, 144, 150, 180 and 216 h. IL-21 stimulated B-cells were harvested after 144 h, and secondary cultures were initiated with irradiated 0.8 × 10^5^ CD40L^+++^-expressing 3T3 cells in the absence or presence of IL-21 for 6, 36 and 72 h. Subsequently, the cells were harvested, stained with LIVE/DEAD Fixable Near-IR and anti-CD19 for 15 min on ice, fixed in 4% paraformaldehyde (Sigma Aldrich, St. Louis, MO, USA) for 10 min at 37 °C and permeabilized with ice-cold methanol (90%) for 30 min on ice. The cells were then stained with BD Phosflow anti-pSTAT3 (clone 4/*P*-STAT3) for 30 min at room temperature. Samples were measured on FACSLSRII and analyzed using FlowJo software version 10 (Treestar, Ashland, OR, USA).

### 2.6. IgM and IgG ELISA of Culture Supernatants

IgM and IgG levels in supernatants were measured as previously described [30]. In short, plates were coated with monoclonal anti-IgM or anti-IgG (2 µg/mL; clone MH15-1 and MH16-1, respectively; Sanquin Reagents, Amsterdam, the Netherlands) and for detection, horseradish peroxidase-conjugated mouse-anti-human-IgM or mouse-anti-human-IgG (1 μg/mL in HPE; clone MH15-1 and MH16-1, respectively; Sanquin Reagents, Amsterdam, the Netherlands) were used. The ELISA was developed with 100 μg/mL tetramethylbenzidine in 0.11 mol/L sodium acetate (pH 5.5) containing 0.003% (*v/v*) H_2_O_2_. The reaction was stopped with 2 M H_2_SO_4_. Absorption at 450 and 540 nm was measured with a Synergy 2 microplate reader (Biotek, Winooski, VT, USA). Results were related to a titration curve of a serum pool in each plate. The lower-limit detection levels of IgM and IgG ELISA were 5 ng/mL and 2 ng/mL, respectively.

### 2.7. 3T3 Cell-Specific BCR/Antibody Assay

3T3 WT cells were harvested, and 2 × 10^5^ 3T3 cells were incubated with culture supernatants that contain secreted antibodies of unswitched (IgM) and class-switched (IgG) isotypes or with 4 µg/mL normal human immunoglobulin (Nanogram 5%; Sanquin Reagents, Amsterdam, the Netherlands) for 30 min at 4 °C in the dark. After incubation, cells were washed with PBS supplemented with 1% bovine serum albumin and subsequently incubated with mouse anti-human IgM (clone MH15-1; Sanquin Reagents, Amsterdam, The Netherlands) and mouse anti-human IgG (clone MH16-1; Sanquin Reagents, Amsterdam, The Netherlands) for 30 min at 4 °C in the dark. Samples were measured on a FACSCanto (BD Bioscience, Franklin Lakes, NJ, USA) and analyzed using FlowJo software version 10 (Treestar, Ashland, OR, USA).

### 2.8. Wide-Field Microscopy

WT and human CD40L-expressing 3T3 cells were harvested and irradiated with 30 Gy. CD40L^+++^-expressing 3T3 cells were labeled with 5.88 μM PKH26 (PKH26 Fluorescent Cell Linker Mini Kit for General Cell Membrane Labeling; Sigma Aldrich, St. Louis, MO, USA) in Diluent C for 5 min at room temperature. Labeling was stopped by adding B-cell medium. Equal numbers (2.1 × 10^4^) WT or CD40L^+++^-expressing 3T3 cells or, alternatively, 9:1 WT to CD40L^+++^-expressing 3T3 cell ratio were seeded in B-cell medium on Nunc Lab-Tek II Chamber Slide 8 well glass slides (Thermo Fisher Scientific, Waltham, MA, USA) to allow adherence overnight. The next day 5.2 × 10^4^ naive B-cells labeled with 0.5 μM CFSE (Invitrogen, Carlsbad, CA, USA) were added and imaged every 10 min for 48 h on a Zeiss Observer Z1.

### 2.9. Real-Time Semi-Quantitative RT–PCR

Different B-cell subsets (as indicated) were sorted. After sorting, RT–PCR was performed as described before [30]. Primers were developed to span exon–intron junctions and then validated (Appendix A). Gene expression levels were measured in duplicate reactions for each sample in StepOnePlus (Applied Biosystems, Foster City, CA, USA) using the SYBR green method (Applied Biosystems, Foster City, CA, USA).

### 2.10. Statistical Analysis

Statistical analysis was performed using Prism 7 (GraphPad, San Diego, CA, USA). The statistical tests used are indicated in figure descriptions. Differences were considered statistically significant when *p* ≤ 0.05.

## 3. Results

### 3.1. The Level of CD40 Costimulation Cooperates with IL-21 and/or IL-4 Signaling to Regulate Human Naive B Cell Expansion and IgG Isotype Switching

The first step in establishing an in vitro human naive B-cell differentiation system was to investigate how variation in CD40 costimulation regulates naive B-cell responses as graded levels of CD40-CD40L interactions regulated expansion and differentiation of total B-cells in vitro [26]. Therefore, 3T3 mouse fibroblasts expressing varying levels of human CD40L were generated (Figure 1A). CD19^+^CD27^−^IgG^−^ human naive B-cells were cultured on WT (=CD40L-negative) or CD40L-expressing cells. The Tfh cell-associated cytokines IL-21 and/or IL-4 were either or not added as it has been shown that in vivo Tfh cells extinguish IL-21 production to switch to IL-4 production, including a transitory IL-21^+^/IL-4^+^ double-positive phase that allowed secretion of either one [7]. In line with a previous report [31], the expansion of naive B-cells was highly reliant on costimulation by CD40L (Figure 1B and Appendix A). Although the proliferation of the naive B-cell population was detected upon CD40 ligation, the number of living cells retrieved 6 days after culture remained low (Figure 1C). This indicates that CD40 ligation is essential for naive B-cell proliferation but that the level of expansion requires additional signals. Indeed, especially in the presence of IL-21, higher CD40 costimulation enhanced B-cell numbers and B-cell proliferation after 6 days in culture (Figure 1B,C and Appendix A). The addition of IL-4 to the culture system supported CD40-mediated B-cell expansion but provided a threshold for the maximum level of CD40 stimulation that could support B-cell proliferation and live B-cell numbers (Figure 1B,C and Appendix A). These data indicate that, especially in the presence of IL-21, costimulation via CD40 controls maximal expansion and maintenance of human naive B-cells in an intensity-dependent manner in vitro.

Analysis of effects of the levels of CD40 costimulation on B-cell differentiation showed that the level of CD40 costimulation positively correlated with BCR class switch recombination (CSR) from IgM to IgG in the presence of IL-21 after 6 days of culture (Figure 1B,D). Again, the addition of IL-4 in the system introduced a maximum threshold for the positive effect of CD40 costimulation, although IL-4 did enhance CSR in conditions where low and intermediate levels of CD40L costimulation were provided than IL-21 alone (Figure 1D). Human BCRs did not recognize 3T3 cell antigens (Appendix A), neither did BCR-ligation using soluble anti-IgM antibodies or anti-IgM-coated particles further affect BCR class switching induced by CD40 ligation and IL-21 and/or IL-4 signaling (Appendix A). Varying levels of CD40 costimulation demonstrated that secretion of IgM and IgG antibodies depended on the strength of costimulation, with the highest level of costimulation being especially supportive for IgM and IgG secretion in the presence of IL-21 (Figure 1E,F). Interestingly, the introduction of IL-4 in the system downmodulated IgM secretion at high CD40 costimulation, but the less affecting secretion of IgG (Figure 1E,F). Correlation between frequency of IgG cells on day 6 and secreted IgG on day 11 was moderate (Appendix A), indicating that part, but not all IgG cells after six days of culture, will survive and differentiate between secreting IgG antibodies. These data indicate that in vitro-induced BCR isotype switching from IgM to IgG and immunoglobulin secretion is determined by the level of CD40 costimulation in the presence of a supporting cytokine environment.

As an alternative to our in-house generated cell lines with variable expression of human CD40L, we cultured human naive B-cells on varying numbers of NIH3T3 cells expressing high CD40L levels [27]. Live imaging of these in vitro cultures demonstrated that the number of B-cells that interact with a CD40L-expressing cell significantly increased when only limited CD40L-expressing cells are available (Appendix A). This suggests that in vitro B-cells only generate stable interaction with cells expressing CD40L. Similar to stimulation of naive B-cells with feeder cells expressing variable levels of CD40L per cell, the proliferation of naive B-cells was not affected by a reduced number of CD40L-expressing cells (Appendix A) but was increased by both cytokines IL-21 and IL-4 (Appendix A). Confirming the observations using cell lines with different expression levels of CD40L, reducing the number of CD40L-expressing feeder cells also altered the number of viable B-cells after 6 days of culture (Figure 1G,H). In IL-4 conditions, B-cell expansion and maintenance were enhanced when the number of CD40L-expressing feeder cells was reduced 9-fold, whereas, in the presence of IL-21 varying, the number of CD40L-expressing feeders did not affect B-cell expansion and maintenance (Figure 1G,H). BCR isotype switching mainly occurred with highly abundant CD40L-expressing feeder cells in the presence of IL-21 (Figure 1G,I). Like stimulations with feeder cells with low and intermediate expression levels of CD40L, the frequency of BCR isotype switching to IgG cells was increased with the lower number of CD40L-expressing cells in the presence of IL-4. Isotype switching is regulated by activation-induced cytidine deaminase (*AICDA)* [32]. Expression of *AICDA* was increased by increasing the number of CD40L-expressing cells in the presence of IL-21 and/or IL-4, with a maximum of *AICDA* expression when only IL-21 was present (Appendix A). While IgM and IgG antibody secretion increased with higher amounts of CD40L-expressing cells and IL-21, IgG antibody-secretion in a micro-environment that included IL-4 showed a maximal beneficial effect of the number of CD40L-expressing cells (Figure 1J,K). This was in line with the observations made when naive B-cells were stimulated with feeder cells exhibiting varying expression levels of CD40L (Figure 1E,F). Altogether these data demonstrate that CD40L expression levels and the number of CD40L-expressing cells both regulate naive B-cell survival, IgG isotype switching, and immunoglobulin secretion.

### 3.2. The Level of CD40 Costimulation in Crosstalk with IL-4/IL-21 Signaling Regulates Differentiation of Human Naive B Cells into CD27^+^CD38^−^ B Cells and Antibody-Secreting Cells

To investigate how variation in CD40 costimulation regulates naive B-cell differentiation into CD27^+^CD38^−^ B-cells, a B-cells phenotype observed in human MBCs in blood and in activated B-cells, and CD27^+^CD38^+^ antibody-secreting cells (ASCs), human naive B-cells were cultured on 3T3s expressing varying levels of human CD40L for 11 days. Naive B-cell differentiation into CD27^+^CD38^−^ B-cells was highly induced in conditions that included IL-4 and low CD40 costimulation (Figure 2A,B). Strikingly, increased levels of CD40L costimulation decreased the formation of the CD27^+^CD38^−^ B-cell population (Figure 2B). IL-21 repressed differentiation into CD27^+^CD38^−^ B-cells in the presence of IL-4 and a low-level of CD40 costimulation (Figure 2A,B). Analysis of CD27^+^CD38^+^ ASC differentiation showed a significant induction, although frequencies were low (to around 3%), and only in the presence of IL-21 (Figure 2C). Interestingly, additional B-cell receptor (BCR) triggering did not affect this differentiation process in vitro (Appendix A).

Next, we assessed whether B-cell differentiation was regulated by the amount of CD40L-expressing feeder cells. Interestingly, the number of CD40L-expressing cells attenuated IL-4-dependent induction of CD27^+^CD38^−^ B-cells, which was decreased by IL-21 (Figure 2D,E). Similar to stimulations with feeder cells that exhibit variation in CD40L surface densities, ASC differentiation occurred to a minimal extent (~0.5–3%), mainly in the presence of IL-21 with highly abundant CD40L-expressing cells (Figure 2D,F).

These data demonstrate that specifically lower CD40 costimulation in the presence of IL-4 supports naive B-cell differentiation into CD27^+^CD38^−^ B-cells in vitro, albeit with low numbers of live CD19^+^ cells. In contrast, higher CD40 costimulation in the presence of IL-21 promotes naive B-cell differentiation into CD27^+^CD38^+^ ASCs.

### 3.3. High CD40 Costimulation in the Presence of IL-21 Induces PRDM1 (BLIMP1) Expression, but Does Not Extinguish B Cell Lineage Program

To elucidate why only limited ASC differentiation was induced after 11 days of culture in the presence of highly abundant CD40L and IL-21, expression of transcriptional regulators of the B-cell and ASC fate was assessed at day 9 (Figure 3A). Comparing FACS-sorted differentiated (CD27^+^CD38^−^) and undifferentiated (CD27^−^CD38^−^) cells after 9 days in culture (Figure 3B) showed that CD40L-expressing cells in the presence of IL-21 significantly induced *PRDM1* mRNA levels in the differentiated subsets, whereas the addition of IL-4 fully abolished *PRDM1* expression (Figure 3C). In contrast, *XBP1*, critical for the development of ASCs, was highly induced in conditions with IL-4 and limited CD40L-expressing cells (Figure 3D). Despite significant induction of the ASC regulator *PRDM1,* the B-cell-specific regulators, *PAX5*, *BACH2* and *IRF8*, were not significantly changed in the CD27^+^ subset compared to CD27^−^ subset (Figure 3E–G). These data show that in vitro stimulated naive B-cells that do not show strong ASC formation become transcriptionally primed for ASC differentiation via upregulation of CD40L/IL-21-mediated *PRDM1* gene expression but still maintain the B-cell identity program.

### 3.4. Secondary Cultures Including CD40L/IL-21 Drive Optimal ASC Differentiation

Next, we assessed which essential cues were lacking in our minimal in vitro culture system to promote substantial terminal ASC differentiation upon naive B-cell stimulation. In vivo, differentiation of B-cells is dependent on repeated interactions and signals provided by Tfh cells in mature GCs that are formed 5 to 7 days after immunization [15]. To achieve a closer simulation of the in vivo situation, cells were harvested 6 days after starting the primary culture and re-cultured in a secondary culture with CD40L-expressing feeder cells and cytokines (Figure 4A). Remarkably, re-culturing under high CD40 costimulatory conditions in the presence of IL-21 induced prominent and significantly more CD27^+^CD38^+^ cells than initial culture only (Figure 4B,C). Renewed culture with low CD40 costimulation also induced ASC differentiation in the presence of IL-21 and IL-4 (and significantly increased the number of live CD19^+^ cells; Appendix A), but not with IL-21 alone. The extensive ASC differentiation after initial expansion resulted in a profound population of ASC (Figure 4D). Remarkably, a more detailed analysis revealed an induction of CD138^+^ PCs upon renewed costimulation/cytokine culture during naive B-cell differentiation (Figure 4E,F). Moreover, the secretion of IgM and IgG was profoundly enhanced in the secondary cultures (5 days cumulative) than the primary cultures (11 days cumulative) and was mediated by IL-21 (Figure 4G,H). These data demonstrate that renewed culture with CD40L and the cytokines IL-21 and IL-4 are required to finalize the differentiation of human naive B-cells into CD138^+^ PCs in vitro.

### 3.5. Second Round of In Vitro Stimulation Switches off the B Cell Lineage Program and Induces Rapid Re-Induction of pSTAT3

ASC differentiation is controlled by a complex interplay of multiple interconnecting transcription factors (Figure 5A). In short, differentiation involves induction of gene expression of the ASC regulator *PRDM1,* which subsequently represses various important pathways that define the B-cell lineage (*PAX5, BACH2 and IRF8*). Further characterization of the effects of renewed costimulation and Tfh-cytokine cultures revealed that mRNA levels of ASC-defining transcription factors *PRDM1* were not further increased in the FACS-sorted differentiated cells (CD27^−^CD38^+^ and CD27^+^CD38^+^ ASCs; Figure 5B) than the CD27^−^CD38^−^B-cells (Figure 5B) after five days into the secondary culture (Figure 5C). In contrast, mRNA levels of *XBP1* were significantly higher in CD27^+^CD38^+^ ASCs than in the other populations (Figure 5D). Additionally, compared to naive CD27^−^CD38^−^ cells derived from primary cultures, *PRDM1* mRNA levels were significantly increased, whereas no increase is observed for *XBP1* (Appendix A). Interestingly, the mRNA levels of B-cell-defining transcription factors *PAX5*, *BACH2* and *IRF8* were significantly decreased in both differentiating populations with the lowest expression in CD27^+^CD38^+^ ASCs and already profoundly reduced in the undifferentiated cells from secondary cultures than initial culture only (Figure 5E–G and Appendix A). These data demonstrate that re-culture with CD40 costimulation and IL-21 yields full extinguishment of the B-cell transcriptional program and induction of full ASC differentiation of naive B-cells.

To address further the link between the extracellular signals provided and the induction of a transcriptional program that efficiently induces ASC differentiation, we investigated the kinetics of phosphorylation of STAT3 (pSTAT3; Figure 6A), the signaling molecule positively regulating *PRDM1* expression and consequently negatively regulating the B-cell transcriptional program (Figure 5A). In line with previous data [33], the presence of IL-21 induced a strong expression of pSTAT3. At the start of culture, pSTAT3 levels in the presence of IL-21 steadily increased and reached maximum levels after 3 days, before return to baseline 3 days later (Figure 6B,C). IL-21 signaling in renewed costimulation cultures induced rapid re-phosphorylation of STAT3 and remained high for at least 3 days (Figure 6D,E). CD27^+^CD38^+^ cells were observed in the secondary cultures in the presence of IL-21 starting from 3 days and rose to around 15% after 5 days (Appendix A). These data demonstrate that re-stimulation cultures with CD40L/IL-21 result in rapid re-phosphorylation of STAT3 correlated with efficient naive B-cell differentiation into ASC formation.

## 4. Discussion

In this study, we established an in vitro culture system that could efficiently differentiate human peripheral blood naive B-cells into ASCs. We demonstrate that naive B-cell survival, abundance and differentiation are dependent on CD40L expression levels by the feeder cells used in the in vitro culture system and on the cytokine environment provided. We also show that renewed costimulation greatly enhanced ASC differentiation. Low CD40 costimulation in the presence of IL-4 prominently induces CD27^+^CD38^−^ cells, compared to high costimulation. CD27 is used as a discriminate marker for human MBCs in blood, raising the possibility that our observed CD27^+^CD38^−^ cells represent the development of the MBCs. However, current discussions in the field attribute the appearance of CD27 to the status of activation, especially in vitro, and not necessarily the formation of long-lived MBCs (reviewed in [34]). Therefore, we cannot make any firm conclusions on MBC formation in our system. Survival of human B-cells under high CD40 costimulatory conditions in vitro was superior in the presence of IL-21. Additionally, these stimulatory conditions made human naive B-cells more prone to differentiate into CD27^+^CD38^+^ ASCs even though the magnitude of ASC formation was low. Although *PRDM1* was upregulated in the culture, the B-cell lineage program was unaltered, possibly contributing to the low magnitude of ASC formation in vitro. However, similar observations have been made in mice in a subset of proliferating dark zone GC B-cells that have low expression of BLIMP1 [35]. These GC B-cells were also not fully committed to PC differentiation but may be more prone to differentiation via that route.

During GC reactions in vivo, GC B-cells repeatedly interact over several cycles, with Tfh cells that provide, among others, CD40 costimulation, IL-21 and IL-4 to the interacting B-cells. Therefore, to mimic the in vivo GC reactions, we utilized secondary cultures, including CD40 costimulation and cytokine stimulation succeeding initial stimulation. Renewed CD40 costimulation in the presence of IL-21 prominently induced ASC formation and concomitantly decreased expression of B-cell-defining transcription factors *PAX5*, *BACH2* and *IRF8*. This demonstrates that also in vitro human naive B-cells, after initial proliferation, require renewed Tfh cell signals to transcend a minimal signaling threshold required for effective differentiation into ASCs. We did observe discrepancies between the expression levels of *PRDM1* and *XBP1,* two transcription factors important in ASC differentiation and function. In secondary cultures, *PRDM1* levels were already increased in the undifferentiated CD27^−^CD38^−^ population and remained constant in the differentiated populations, whereas *XBP1* levels only increased in the fully differentiated CD27^+^CD38^+^ ASC. This indicates that *PRDM1* is important before the fate transition to really drive differentiation, whereas *XBP1* is relevant for the large production of antibodies.

Previous studies using murine naive B-cells reported that CD40L stimulation, with or without BAFF stimulation, together with phasic (re-)stimulation of IL-21 and IL-4, resulted in significant expansion and induction of GC-like cells and differentiation into CD138^+^ cells. These cells required an adoptive transfer model to fully mature into long-lived PCs [36,37]. We here show that for human in vitro naive B-cell differentiation, renewed costimulation and Tfh-cytokines during the B-cell priming phase induce potent plasmablast (PB) and PCs formation from CD40L-stimulated naive B-cells in the presence of IL-21 −/+ IL-4. Whether these are long-lived PCs as observed in the adoptive transfer model [37] remains to be determined. Other groups have demonstrated in vitro differentiation of human MBCs into long-lived PCs using an intricate 3-step culture system, in which stromal cell factors, mimicking the bone marrow environment, prolonged the survival of the in vitro-generated PCs [38,39]. This could be the next step in our human naive B-cell culture system to assess the longevity of our induced ASCs.

Renewed CD40 costimulation of human naive B-cells in vitro with high CD40L levels in the presence of IL-21 rapidly re-induced STAT3 activation, an IL-21 receptor signaling protein. Individuals with STAT3 mutations and mice deficient for IL-21 or IL-21 receptor have shown that STAT3 is essential in GC formation and maintenance, PC differentiation and immunoglobulin secretion [18,19,40,41,42]. Furthermore, IL-21 synergizes with CD40L stimulation to induce expression of the ASC-defining transcription factors *PRDM1* and *IRF4* in human B-cells [43]. Altogether, the observed rapid re-induction of STAT3 in the secondary in vitro cultures could explain the more prominent ASC induction and suggests that dynamic reexpression of STAT3 is important in ASC differentiation.

The establishment of this minimalistic in vitro culture system that supports efficient differentiation of human naive B-cells into antibody-secreting cells while also maintaining high cell numbers to investigate the differentiation pathways may form an important step for the much-desired in-depth dissection of human B-cell terminal differentiation. It may eventually prove beneficial in devising new strategies and targets for treating antibody/ASC-mediated inflammatory and auto-immune diseases and might prove invaluable for monitoring vaccination efficiency.

## Figures and Tables

**Figure 1 cells-10-01183-f001:**
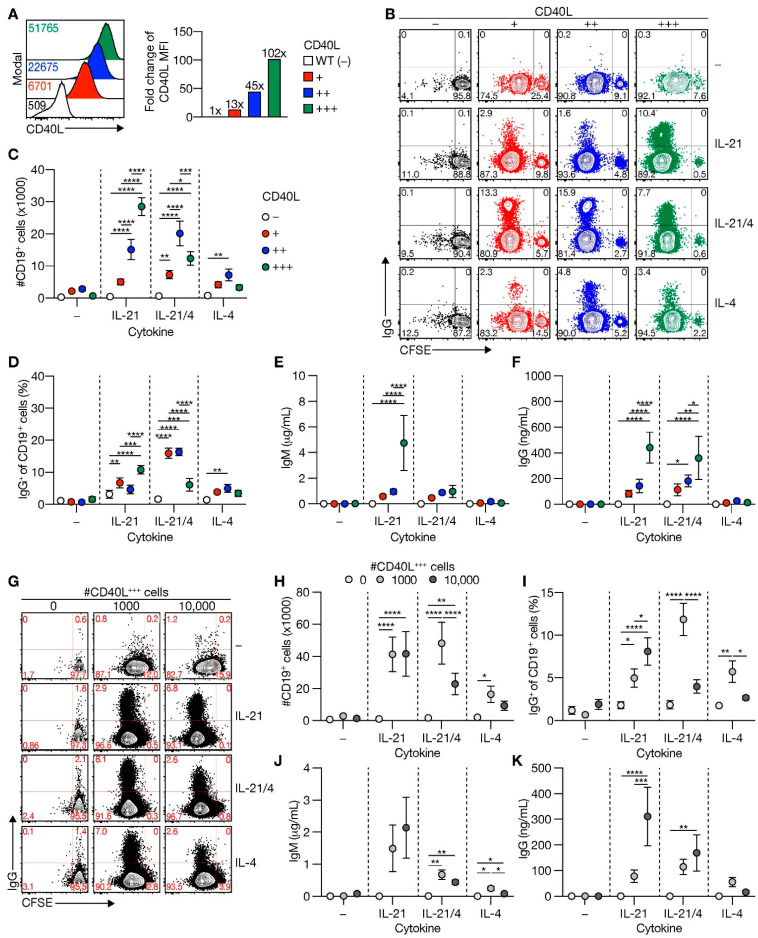
Level of CD40L expression and number of CD40L feeder cells supports human naive B-cell expansion and IgG class switching. (**A**) Representative histograms (left) of human CD40L expression stably transfected 3T3 fibroblasts expressing various amounts of human CD40L (+, ++ or +++) compared to non-transfected 3T3 cells (WT; −). Numbers indicate the mean fluorescence intensity (MFI) of CD40L expression. Fold changes compared to WT controls of CD40L MFI are shown in the right panel (data shown represent three separate experiments). (**B**–**D**) Human CFSE-labeled naive B-cells were cultured on 3T3 cells expressing varying levels of CD40L with or without IL-21 and/or IL-4 for 6 days and analyzed for proliferation and class switching to surface IgG ((**B**); representative plots). The number of live CD19^+^ B-cells (**C**) and surface IgG-expression (**D**) was quantified after the cultures by flow cytometry analysis (*n* = 6). (**E**–**F**) Cumulative secretion of IgM ((**E**); *n* = 9) and IgG ((**F**); *n* = 8) measured in culture supernatants after 11 days. (**G**) Representative plots of CFSE-labeled human naive B-cells cultured on 0/1000/10,000 CD40L^+++^-expressing 3T3 cells (supplemented to 10,000 with WT 3T3 cells) with or without IL-21 and/or IL-4 for 6 days. (**H**–**I**) Number of live CD19^+^ events ((**C**); *n* = 9) and the frequency of IgG^+^ cells ((**D**); *n* = 7) after culture on 0/1000/10,000 CD40L^+++^-expressing 3T3 cells, and cytokines for 6 days. (**J**–**K**) Cumulative IgM ((**F**); *n* = 7) and IgG ((**G**); *n* = 7) levels measured in culture supernatants after 11 days. Data are shown as mean ± SEM of independent experiments. Single experiments were conducted in triplicate. Data were analyzed by a two-way ANOVA followed by Tukey’s multiple comparison test. * *P* ≤ 0.05, ** *P* ≤ 0.01, *** *P* ≤ 0.001, **** *P* ≤ 0.0001.

**Figure 2 cells-10-01183-f002:**
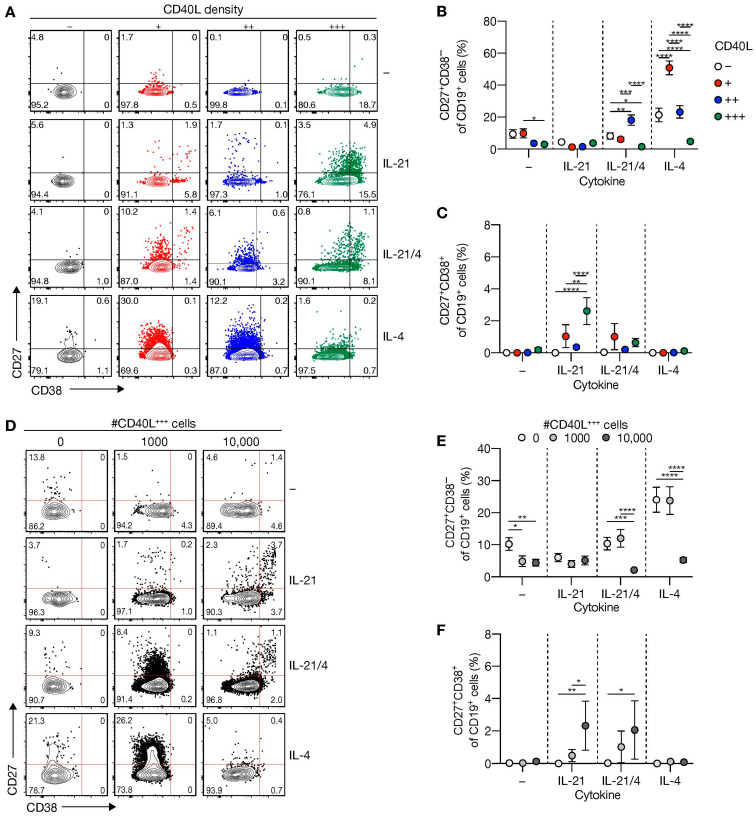
Level of CD40L expression and number of CD40L feeder cells regulates differentiation of human naive B-cell into CD27^+^CD38^−^ B-cells and CD27^+^CD38^+^ antibody-secreting cells. (**A**) Representative of FACS plot showing CD27 and CD38 expression among human naive B-cells cultured on CD40L-expressing 3T3 cells (as in Figure 1A) with or without IL-21 and/or IL-4 for 11 days. (**B**,**C**) The frequency of CD27^+^CD38^−^ (**B**) and CD27^+^CD38^+^ (**C**) populations were analyzed 11 days after culture on CD40L-expressing 3T3 cells and cytokines. (**D**) Representative plots of human naive B-cells cultured on 0/1000/10,000 CD40L^+++^-expressing 3T3 cells (as in Figure 1A) supplemented to 10,000 with WT with or without IL-21 and/or IL-4 for 11 days. (**E,F**) The frequency of CD27^+^CD38^−^ (**E**) and CD27^+^CD38^+^ (**F**) cells 11 days after culture on different numbers of CD40L^+++^-expressing 3T3 cells and cytokines. Data are shown as the mean ± SEM. (*n* = 6 (**A**–**C**) and *n* = 7 (**D**–**F**) independent experiments). Single experiments were conducted in triplicate. Data were analyzed by a two-way ANOVA followed by Tukey’s multiple comparison test. * *P* ≤ 0.05, ** *P* ≤ 0.01, *** *P* ≤ 0.001, **** *P* ≤ 0.0001.

**Figure 3 cells-10-01183-f003:**
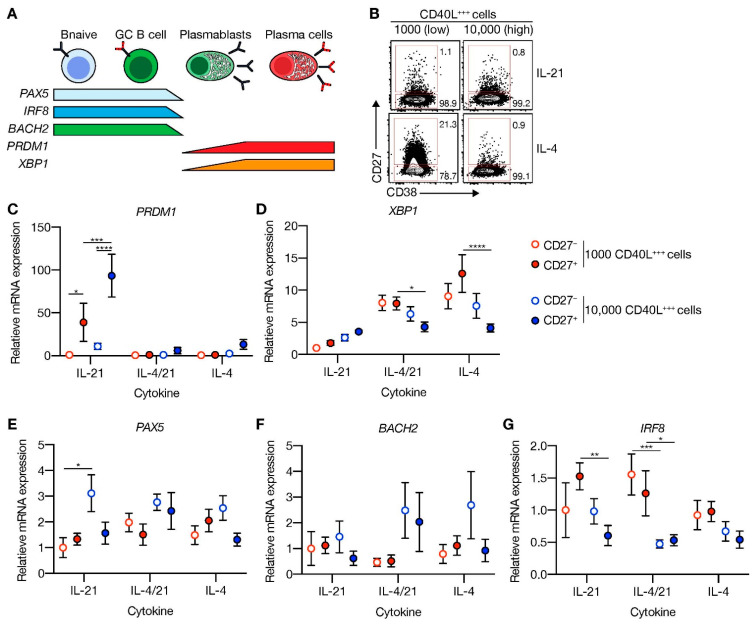
CD40 costimulation together with IL-21 signaling induces PRDM1 expression and does not extinguish the B-cell lineage transcriptional program. (**A**) Schematic overview of the cellular stages and important transcription factors involved in B-cells differentiation from naive to antibody-secreting plasma cell (GC = germinal center). (**B**) Representative plots of human naive B-cells cultured on 1000 (low; supplemented with 9000 WT 3T3 cells) or 10,000 (high) CD40L^+++^-expressing 3T3 cells (as in Figure 1A) with or without IL-21 and/or IL-4 for 9 days. Subsequently, CD27^−^CD38^−^ (CD27^−^) and CD27^+^CD38^−^ (CD27^+^) cells were purified by cell sorting. (**C**–**G**) Expression of *PRMD1* (**C**), *XBP1* (**D**), *PAX5* (**E**), *BACH2* (**F**) and *IRF8* (**G**) mRNA in the sorted populations were analyzed by qPCR and related to levels present in the CD27^−^ cells that were cultured on 1000 (low) CD40L^+++^-expressing 3T3 cells together with IL-21. Data are shown as mean ± SEM. (*n* = 5 independent experiments). Single experiments were conducted in triplicate. Data were analyzed by a two-way ANOVA followed by Tukey’s multiple comparison test. * *P* ≤ 0.05, ** *P* ≤ 0.01, *** *P* ≤ 0.001, **** *P* ≤ 0.0001.

**Figure 4 cells-10-01183-f004:**
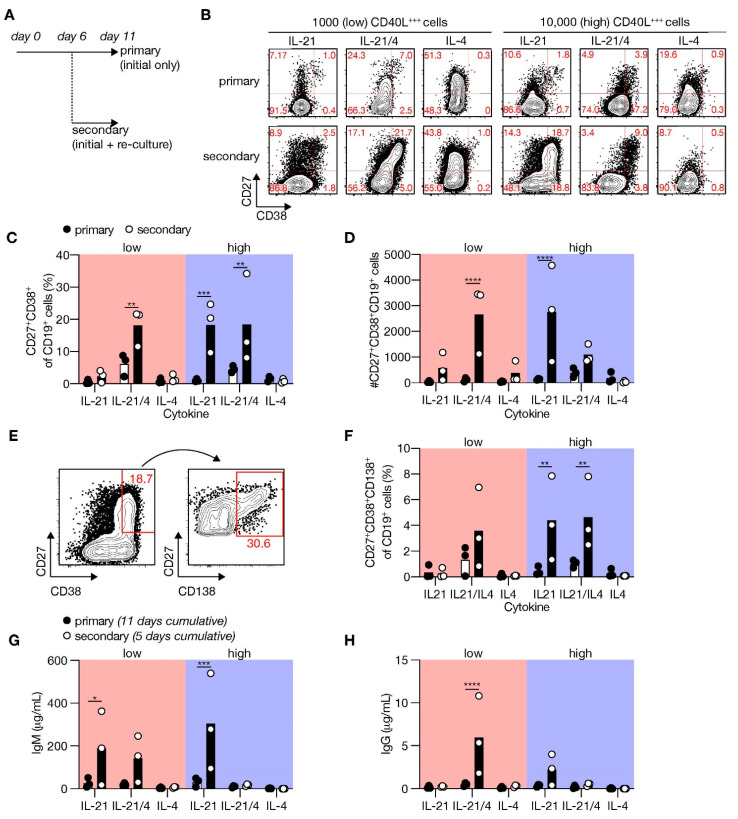
Renewed CD40 costimulation and Tfh cytokines induce in vitro differentiation of human naive B-cells into antibody-secreting cells (ASCs). (**A**) Human B-cells cultured on 1000 (low; supplemented with 9000 WT 3T3 cells) or 10,000 (high) CD40L^+++^-expressing 3T3 cells (as in Figure 1A) with or without IL-21 and/or IL-4 for 11 days (primary—initial only stimulation). Alternatively, primary cultures were harvested after 6 days, and secondary cultures were initiated for 5 days with the same number of CD40L^+++^-expressing 3T3 cells and similar cytokine environments as in the primary culture. (**B**) Representative of FACS plots showing CD27 and CD38 expression among human B-cells cultured as in A. (**C**,**D**) The frequency (**C**) and number (**D**) of CD27^+^CD38^+^ cells were analyzed 11 days after initial only, or initial + secondary culture. (**E**) Representative plots of CD138 expression within the CD27^+^CD28^+^ ASC population. (**F**) The frequencies of CD138^+^ plasma cells were determined 11 days after initial only or initial + secondary culture. (**G**,**H**) Cumulative secretion of IgM (**G**) and IgG (**H**) measured in culture supernatants 11 days after initial only or 5 days after initial + secondary culture. Each data point represents the mean of an individual experiment (*n* = 3) with triplicate measurements. Mean values are represented as bars. *p*-values were calculated using multiple t-tests. * *P* ≤ 0.05, ** *P* ≤ 0.01, *** *P* ≤ 0.001, **** *P* ≤ 0.0001.

**Figure 5 cells-10-01183-f005:**
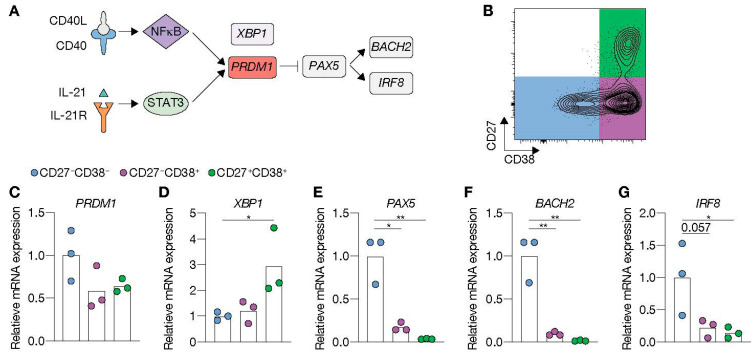
Renewed CD40 costimulation and IL-21 signaling drives antibody-secreting cell (ASC) differentiation and represses transcriptional program related to the B-cell fate. (**A**) Schematic representation of BLIMP1 activation upon CD40 and IL-21 receptor (IL-21R) signaling. STAT3, together with NFκB, regulates BLIMP1 expression. BLIMP1 downregulates *PAX5* expression and, consequently, its downstream targets *BACH2* and *IRF8*. (**B**–**G**) Human naive B-cells were cultured on 10,000 (high) CD40L^+++^-expressing 3T3 cells (as in Figure 1A) for 6 days with IL-21. After 6 days, secondary cultures were initiated for 5 days with the initial stimuli. (**B**) Subsequently, CD27^−^CD38^−^, CD27^−^CD38^+^ and CD27^+^CD38^+^ cell populations were sort purified. (**C**–**G**) Expression of *PRMD1* (**C**), *XBP1* (**D**), *PAX5* (**E**), *BACH2* (**F**) and *IRF8* (**G**) mRNA in sorted populations were analyzed by qPCR and related to levels present in CD27^−^CD38^−^ cells. Each data point represents the mean of an individual experiment (*n* = 3) with triplicate measurements. Mean values are represented as bars. *p* values were calculated using RM one-way ANOVA followed by Tukey’s multiple comparison test. * *P* ≤ 0.05, ** *P* ≤ 0.01.

**Figure 6 cells-10-01183-f006:**
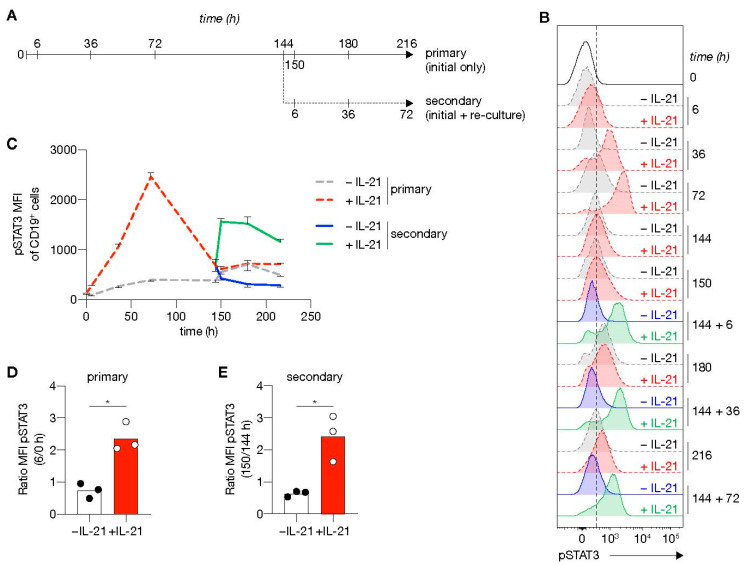
STAT3 is rapidly re-phosphorylated upon re-stimulation using CD40 costimulation and IL-21 signaling. (**A**) Overview of the time points sampled from primary and secondary human B-cell cultures. Human naive B-cells were cultured on 10,000 CD40L^+++^-expressing 3T3 cells (as in Figure 1A) with or without IL-21 for 6 days. After 6 days, secondary cultures were initiated for 3 days with 10,000 CD40L^+++^-expressing 3T3 cells with(out) IL-21 (*n* = 3). (**B**,**C**) Representative plot of pSTAT3 levels (**B**) and quantification (**C**) at baseline and 6, 36, 72, 144, 150, 180 and 216 h in primary culture (dotted gray w/o IL-21 and red lines with IL-21); and 6, 36 and 72 in secondary culture (solid blue w/o IL-21 and green lines with IL-21). Data are shown as mean ± SEM (*n* = 3 independent experiments). Single experiments were conducted in triplicate. (**D**, **E**) Fold induction of pSTAT3 6 h in primary (**D**) and secondary (**E**) cultures relative to time point 0 or 144 h. Each data point represents the mean of an individual experiment (*n* = 3) with triplicate measurements. Mean values are represented as bars. *p* values were calculated using paired t-test. * *P* ≤ 0.05.

## Data Availability

The data presented in this study are available on request from the corresponding author.

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
