# Peer review of "Minimalistic In Vitro Culture to Drive Human Naive B Cell Differentiation into Antibody-Secreting Cells"

_cells, 2021, doi:10.3390/cells10051183_

Round 1

Reviewer 1 Report

In this manuscript, authors described their efforts to determine culture conditions for efficient proliferation, class switching and plasma-cell differentiation of human B cells in vitro. The determined conditions of a level of CD40L-expression on feeder cells and of cytokines will be a useful information for those who wish to expand human B cells and produce IgG antibodies from such cells. Particularly valuable is their finding that the secondary culture with the feeder cells and IL-21 with/without IL-4 efficiently induced plasma-cell differentiation with down-regulation of expression of PAX5, BACH2 and IRF8.

In this regard, authors should show actual number of CD27+ CD38+ plasma(blast) cells in Figure 4C, not just percentages, so that researchers can apply this protocol to get maximum number of human antibody-forming cells.

In addition, they should show the isotypes and IgG subclasses of the plasma(blast) cells induced by the secondary culture; it would be useless for some purpose if they are mostly of IgE, for example.

Minor comments:

  1. Page 2, line 83: please specify the animal species from which the CD40L cDNA was derived.
  2. Page 2, line 92: “Compared to our transfectants, NIH3T3 fibroblasts expressed high (CD40L+++) human CD40L levels [27]”: What does this sentence exactly mean?
  3. Page 6, line 267-268: “antibody-secretion in a micro-environment that included IL-4 showed a maximal beneficial effect of the number of CD40L-expressing cells (Figure S3F-G)”: Meaning is unclear. Is this true for IgM (Figure S3F)?
  4. Page 6, line 275: In this subtitle, the statement “… differentiation of human naive B cells into memory B cells …” is inappropriate since the CD27+ CD38+ cells are not necessary MBCs but more likely be just activated B cells in culture, as authors themselves stated in Discussion (page 11, line 422-424).
  5. Page 9, line 354-356: The reference “(Figure 4E-F)” is incorrect.
  6. Page 10, line 403-405: A comment by a former reviewer?
  7. Supplementary Figure S5B: Labelling at the vertical axis “CD38-/+” should be as “CD38-“.

Reviewer 2 Report

The manuscript from Unger et al. reports a novel methodological approach to understand how naïve B lymphocytes differentiate into antibody-secreting cells (ASC). Through an interesting experimental set, they use a co-culture system of CD19+ CD27- IgG- B cells with murine 3T3 fibroblasts, expressing different levels of CD40L. With this method, together with the co-stimulation with IL-21 and IL-4 (alone or in combination), they simulate in vitro the interplay between naïve B cells and T follicular helper (Tfh) lymphocytes, that takes place in germinal centers (GC), mediated by CD40L and its receptor, CD40.

The manuscript is written in a clear manner and the experimental setting is exposed in a detailed way, facilitating the comprehension to the reader. I believe that the method presented here can be of great utility to improve the already existing approaches to study GC reaction and ASC. However, this apparent strength of the study turns into a major concern, since the resulting data does not contribute with any further novelty to the already existing knowledge in the field. The authors use the experimental setting to dissect how terminal B cell differentiation depends on a very tight regulation of CD40L levels and on which cytokines are used to co-stimulate naïve B cells, but the final differentiation of cells into ASC has been widely reported previously (Lesley et al. PNAS 2006; Robinson et al. Immunol Cell Biol 2019; Nojima et al. Nat Comms 2011). In fact, I believe that the last paragraph of the discussion (lines 472-478) contains the most relevant conclusion of the work which, unfortunately, is not reflected in the title and only scarcely in the abstract. I would suggest to “move the focus” and to re-direct both title and abstract of the manuscript towards the relevance of the in vitro system described, rather than focusing on the resulting data derived from it.

Aside from these general comments, specific concerns are listed point-by-point below.

Major points:

  1. In Figures 1B and 1F, it seems that naïve B cells in presence of CD40L (at any of the concentrations used and both using IL-4 and IL-21), transit to more differentiated cells that express IgG. These results do not correlate with that presented in Figure 3, where cells treated with the highest amount of CD40L and IL-21 display a strong induction of PRDM1, whereas in the other conditions, there is no effect on PRDM1 In Figure 1B, CD40L+++ cells induce around 10% population of IgG+ cells when treated with IL-21. The addition of IL-4 only reduces the % of IgG+ population to 7.7%. Is this enough to explain a total abrogation of PRDM1 (typical marker of ASCs)?
  2. It can be extracted that one of the reasons to explain that a single stimulation with CD40L and IL-21 is not sufficient for a complete ASC differentiation is the Spearman value mentioned in line 239, referring to the correlation between IgG expressing cells (at day 6) and the amount of IgG secreted (at day 11). The authors consider this correlation (0.47) as low. However, Spearman values vary from -1 to +1 and a positive correlation around 0.5 is generally considered as moderate (ranging from 0.40 to 0.59). These data should be shown and further explained why the authors consider that this correlation is too low.
  3. In relation to the previous points, in Figure 2A, there is a 6% of CD27+ CD38+ population in the CD40L+++ condition treated with IL-21 alone. Instead, in Figure 3B, the percentage of CD38+ cells is almost negligible in both conditions of CD40L stimulation. How do the authors explain that? I guess that a possible answer is that data in Figures 1 and 2 is obtained after 11 days of stimulation and in Figure 3, the results show the effect of a 9-day treatment. In this case, and assuming that such a great effect can occur only in 48h, it is tempting to understand that the complete terminal B cell differentiation does not depend on a secondary stimulation with CD40L (as shown in Figures 4-6), but it is just a matter of time and that a single CD40L + IL-21 stimuli can complete differentiation of naïve B cells into ASCs in a longer period of time.
  4. The experimental setting in Supplementary Figure S3 is not fully convincing. The authors quantify the number of cells that interact with each single CD40L+++-secreting 3T3 fibroblast. It can be expected that naïve B cells search CD40L stimulation and, therefore, they interact with the modified cells expressing it (and not with the WT ones). This point should be considered by the authors and the conclusions extracted should be rephrased.
  5. Moreover, for undetermined reason, the experimental setting used in Figures 1 and 2 (with different subclones of 3T3 expressing varying amounts of CD40L) is replaced by a more complex setting, in which naïve B cells are seeded in presence of 10000 CD40L+++ 3T3 cells or only 1000 of these cells (supplemented with 9000 WT cells). It is ok to use this setting to confirm the data, but I believe that it could be just paralleled by comparing CD40L+ and CD40L+++ 3T3 cells. Why do the authors change the experimental setting in Figure 3 and maintain it until the end of the manuscript? At least some of the experiments should be reproduced using CD40L+ subclone cells instead of the condition with 1000 CD40L+++ cells.
  6. Lines 263-265: Referring to Supplementary Figure S3, the authors mention that AICDA expression is increased upon addition of IL-4 to IL-21 co-stimulation. However, the maximum expression of AICDA occurs under treatment with IL-21 alone, in the condition of highest CD40L levels. This should be corrected.
  7. The use of CD138 marker has been widely reported to identify plasma cells (McCarron et al. Blood 2017). The authors only start using this marker in Figure 4 of the manuscript. Despite the use of CD38 as marker of plasma cells is also accepted (Alessio et al. J Immunol 1990), the authors employ CD138 as a marker of more mature plasma cells, with capacity to secrete antibodies. However, they only use this marker in the context of a secondary stimulation with CD40L (such as in Figure 4D). The same analysis from Figure 4D should be reproduced for the CD27+ CD38+ population in Figure 2A (in the CD40L+++ with IL-21 condition), to corroborate whether the differentiation of naïve B cells into ASC is only completed after a renewed CD40L stimulation (considering CD138 as ultimate marker of this process).
  8. Figures 6B and 6C are not convincing. They use pSTAT3 as readout of CD40 stimulation with CD40L and IL-21.
  9. First, the decrease of phosphorylated STAT3 between 72 and 144h after primary stimulation is too drastic and it is probably due to a lack of intermediate time points. Some intermediate points (e.g. 96 or 120 hours) should be included.
  10. The tendence of pSTAT3 levels 72 hours after a second stimulation seems to be very similar to the one observed after the primary CD40 induction. Therefore, further time points beyond 72h (again, maybe 96 or 120 hours) should be collected.

Minor points:

  1. In Figure 1D, it is mentioned that IL-4 enhances CSR in conditions of low and intermediate levels of CD40L. In the Figure, it seems that the effect of IL-4 only takes place in absence of IL-21. This effect should be clarified.
  2. Figures 2B-2C: The chart legend is missing and, therefore, the use of colours to label each condition is not clear.
  3. The use of Supplementary Figures is confusing. For instance, Suppl. Figure S4 is mentioned before some panels of Suppl. Figure S3 (lines 252-259) and some panels of Suppl. Figure S2 are mentioned after Suppl. Figures S3 and S4 (line 293). This should be reorganized because it is confusing for the reader.
  4. Supplementary Figure S3A: Images should be more similar between them to compare. Probably, using both images in a bright field setting would be a better choice.
  5. Figures 3C-3G: The colors used to differentiate between CD27+ and CD27- populations are too similar and can be confusing. They should be modified.
  6. Figures 3E-3G: The scale of Y axis of these charts is too high. It is not necessary to show the Y axis up to 5 or 10. Just scaling up to 2-3 (depending on the case) would be enough. As it is now, the charts provide a false sensation of no changes in expression of these genes between the different conditions.
  7. Line 353: Figure S6A should be named as S6, since there is only one panel.
  8. Line 356: Figures 4E-F refers to Figures 4D-E. It should be corrected.
  9. Lines 403-405: These lines should be deleted, since they probably belong to a template form provided by the journal.
  10. Supplementary Table should be named as S1 and not as S2, since there is only one supplementary table.
  11. The format of references should be checked. The name of authors in reference 6 is written in capital letters.

Reviewer 3 Report

Unger et al. investigated "minimal conditions," including extent of CD40L and IL-21 and IL-4 stimulation, which coax human naive B cells to differentiate into plasma cells. This subject has always attracted interest for theoretical reasons but recently the subject capture interest because differentiation of selected human antibody-secreting cells potentially facilitates production of human antibodies for therapeutic uses. The experimental approach, including the testing of phenotype and Ab production after stimulation of naive B cells in culture, appears capably executed and the results generally confirm what others find using similar approaches. The work takes a few novel turns but does so superficially as the manuscript in present form provides little new information. However, more focused and detailed consideration of novel aspects of the work could make an estimable contribution.

Specific comments:

  1. One original, or potentially original, aspect of the work concerns the stimulation of B cells with 3T3 fibroblasts expressing various levels of CD154 (CD40L). Although others reported human B cell responses to murine cells expressing CD154, but Unger et al. explored the impact of variations in expression - "absolute" levels of expression and more importantly the possibility that steric factors, i.e. competition for CD154 could influence stimulation. This distinction could be quite important in lymphoid tissues, which the authors believe they model, but competition for access is not precisely and critically explored. Hence, the conclusions mainly re-state the experimental conditions used, for example "renewed CD40 ligation" which refers to the removing of B cells from initial cultures and re-plating CD40-3T3 cells etc. The authors could have devised conditions that would distinguish dynamic CD40-CD154 interaction (on and off versus some fraction being persistently on) of individual cells cells versus competition for CD154 expressing cells (some cells are on and some off) but they did not.

  1. Nor do the authors verify that CD40-CD40L interactions rather than some other variable determine key results. Where on 3T3 cells is CD40L expressed (apical? basal? uniformly?) and how were levels determined? That is, where cells harvested or was expression tested using adherent cells-the method needs to be completely stated. If cells were harvested, were levels confirmed using adherent cells? Since CD40-CD40L is the key interaction in this work the authors should consider using blocking Ab to confirm some findings.

  1. The authors, like too many others, do not consider whether B cell antigen receptors (BCR) were stimulated, during isolation or during culture with heterologous cells or both. Remarkably, the authors make no effort to explain why experiments depicted only in the supplement lead to the conclusion that: "BCR stimulation does not naive B cell differentiation upon CD40 co-stimulation" (the experiments leading to this conclusion are superficial and incomplete). Perhaps BCR was already stimulated. Some fraction of human BCR recognizes 3T3 cells. Do the culture conditions select these B cells? Do the B cells adhere to 3T3 cells (as Figure S3 suggests) because of these BCR? Do the results ascribed to "renewed CD40 ligation" actually reflect an increase in the fraction of B cells with stimulated BCR or is the competition for CD40 actually competition for antigen?

  1. A more incisive and original study would have included some effort to explore what the antibodies recognize. If all of the naive B cells are truly naive, all might recognize distinct antigens and hence there might be no ascertainable specificity. But, if the experiments selected B cells specific for 3T3 then perhaps the supernatants would contain anti-3T3 Ab. Were the Ab polyreactive? One appreciates the amounts of Ab are small but the same method used to detect IgG and IgM could have been used to test specificity. Further to this point, if anti-3T3 Ab were to be produced, the Ab might be cleared by the irradiated cells (we are never told how long these cells survived in the cultures). This question is important as the impact ascribed to cytokines could also reflect the impact on fibroblast survival.

  1. The selection of results for illustration in the manuscript and for relegation to the supplement is certainly odd. Many of the images depicting analysis of cytokine responses are confirmatory at best and distract from novel results at worst. I suggest the authors minimize presentation in the main text and move these figures to the supplement. Responses to BCR ligation should have included cells stimulated and co-stimulated in the absence of 3T3 cells and should have been presented and explained since these speak to the experimental system used. Likewise, relevant parts of Figure S3, S5 and S6 should be included in the manuscript so readers do not need to consult the supplement to grasp the main points.

  1. A few minor points are as follows: The last paragraph of Results should be deleted. Mention of "in vivo" in the abstract sets a false expectation as no in vivo verification is or can be provided. The methods should clearly indicate: (a) whether and how long B cells were rested before introduction into culture; (b) whether and when medium was changed; (c) what is the control when CD40L stimulation is "renewed" and does the control receive "fresh medium."

Round 2

Reviewer 2 Report

In this revised version, the manuscript from Unger et al. has improved in some aspects, clarifying the use of the techniques involved and emphasizing the importance of the experimental setting, rather than the results obtained themselves.

However, most of the major points have not been addressed as requested. In fact, this revised version has been submitted in only two weeks. I do not understand the reason of resubmitting the manuscript in such a short period of time but, given that some of the in vitro models used require around 10-12 days, it is obvious that the new data requested has not been included. Therefore, despite enhancing some aspects of the text, the overall quality of the manuscript has not been improved.

Some specific examples are detailed:

  • Changes in the title and the abstract

I appreciate the intention of the authors in modifying the title and the abstract, but that was not the point of my comments. I did not mean just to include the words “in vitro” in the title. The title is still giving importance to the fact that CD40L and IL-21 induce differentiation of B cells into ASC, which is already known.

  • Response to Issue #2

Where do these new data points come from? Did the authors perform these new experiments?

At this point, it is clear that not all IgG-producing cells will eventually become ASC. The problem here is that the authors have not modified their conclusions. The authors accept that the correlation is moderate, but they maintain the same conclusions than before, which were raised under the premise of a low correlation.

  • Response to Issue #3

First of all, I do not think that the standard error of the mean (s.e.m.) in the IL-21 condition of Figure 2C is that large as the authors comment. I believe it is quite acceptable and, in fact, there are significant differences with other conditions. In consequence, I think that the donor included in Figure 2A is not representative of the whole cohort of donors, since it has a 6% of CD27+ CD38+ cells. Otherwise, are the samples included in Figure 3 the representative ones? Or should I rely on data in Figure 2?

I understand the explanation and technical problems related to differential behavior of samples derived from distinct individuals are totally acceptable. The issue is that the Figures shown in the manuscript present discrepancies and that, even after warning the authors on that point, they have not made any changes to avoid or address these discrepancies, affecting the message transmitted to the reader.

  • Response to Issue #5

I appreciate that the authors care about how plausible it will be for other laboratories to reproduce the data they present, and it is worth mentioning this point in the manuscript. Nevertheless, this is not the point of my concern, which was focused on why the experimental setting in the initial Figures (with 3T3 cells secreting varying gradients of CD40L) is replaced from Figure 3 until the end of the manuscript for a less elegant setting, in which they only modify the number of 3T3 cells plated (all of them with the same level of CD40L secretion).

Even more, the incorporation of Figures that were previously allocated in the supplementary material to the main manuscript reduces the importance of the experimental setting with CD40L expression gradients. There was no need to do that.

  • Response to Issue #6

As it happened before in Issue #2, the authors modify the sentence, but the conclusions of the experiment are maintained. It seems that being aware of misinterpretation of the results does not lead them to modify their conclusions.

  • Response to Issue #7

The authors have not answered to my request of analyzing CD138 expression in the experimental setting of Figure 2A. Specifically, I asked about the percentage of CD138+ cells in the CD27+ CD38+ subset of cells (treated with CD40L+++ 3T3 cells and IL-21). They state that they have analyzed the % of CD138+ cells in primary and secondary cultures used in Figure 4, but that was not the point.

Even more, Figure 4F displays a high variability, which is higher than the variation in Figure 2C, that they use to answer Issue #3. Figure 4F should include further experimental points.

  • Response to Issues #8, #9 and #10

I really appreciate the effort of the authors in trying to decipher the mechanisms underlying B cell differentiation into ASC. In fact, this could be a strength of the paper, but the data is not convincing. The intermediate time points that I requested have not been included. I understand that it may be difficult to obtain samples from the donors but, given that the manuscript has been resubmitted in only two weeks, I think that the authors have not even tried to obtain new data in these assays.

Since the authors have not properly addressed most of the major issues raised, I consider that the manuscript does not merit publication in Cells.

Reviewer 3 Report

The authors have answered most questions and made appropriate revisions.  Still unaddressed is the question of whether the naive human B cells recognize 3T3 cells via B cell antigen receptors (BCR).  The authors believe that "BCR repertoire analysis" (was that by VDJ sequencing?) revealing polyclonality excludes the possibility that the B cells recognize transfected 3T3 cells.  However, if the B cells are naive, then many BCR are polyreactive and hence might recognize an heterologous cell line.  Excluding this possibility experimentally is so simple, the authors should do it - if polyreactive human Ab do not recognize transfected 3T3 cells, the authors will have a nice point to add and will appear more rigorous than others and if the cells are recognized, they will have revealed a critical limitation of some other work.

Round 3

Reviewer 2 Report

After a second round of review, I appreciate that the authors have incorporated some of the modifications that were suggested. In fact, some of them were already suggested in the initial round of review and have not been included until this second round.

However, after these two rounds, there has not been a substantial change in the content of the article and most of the experiments suggested to improve the quality of the manuscript have not been conducted. The first round of comments was replied in less than two weeks and this second round, in less than one month. I understand that some experiments can not be performed in such a short amount of time, but there was no need to reply in less than one month. The authors should have probably taken more time to properly reply to the comments and to incorporate the experiments required in an appropriate manner.

Detailed comments of each specific issue are listed below:

Response to Issue #1

This new title is more appropriate than the previous one, since it reflects better the strongest point of the paper, which is the in vitro culture system developed, rather than the results obtained.

Response to Issue #2

Ok, I understand that the data can be obtained from already ongoing assays in the lab. Therefore, it is even hard to understand why the authors have not included any further assays in other issues raised (as it will be discussed later). Regarding the conclusion raised, the modification is correct, but it does not provide much novelty, since it is known that not all IgG cells differentiate into antibody-secreting cells.

Response to Issue #3

Thanks to the explanation, I understand better the difference between these Figures. However, they have maintained the same donor in IL-21 condition of Figure 2A, which is not representative of the whole assay. It should have been replaced.

Response to Issue #5

After two rounds of review, the authors have not replied to why they replace an experimental setting for another one. From my point of view, the setting used in the initial Figures (1 and 2) is more elegant than the other setting in which they modify the number of CD40L-secreting cells (only using CD40L+++ cells). The response of the authors is only focused in whether they move figures from supplementary material to the main text or not, but they not even reply to the main point of the comment.

Response to Issue #6

The new sentences incorporated here reflect the results obtained in a more adequate manner.

Response to Issue #7

The figure shown in the cover letter is not convincing. They have only analyzed the % of CD138+ cells in 3 assays (there are only 3 points in each condition). I believe that they should incorporate this Figure in the manuscript, but with further data points. As it is now, it is not convincing at all. Given that it seems that these experiments are extensively carried out in the lab continuously (as they state in the answer to Issue #2), they should have further data to analyze the presence of CD138+ cells.

Response to Issues #8, #9 and #10

After two rounds of review, no modifications have been introduced to Figure 6. I can understand that the material (human B cells) is limited. However, they mention in response to Issue #2 that the culture conditions are used extensively in the lab. Maybe they can not cover the whole range of time points required but, at least, I would have highly appreciated that they try to include some further points (intermediate points in the primary culture and longer time points in the secondary culture).

In summary, since the quality and the novelty of the manuscript have not increased in a significant manner since the initial version, I maintain my decision that it does not merit to be published in Cells.
